# A Review of the Classification of Opal with Reference to Recent New Localities

**Neville J. Curtis [1,2]**, **Jason R. Gascooke [2]**, **Martin R. Johnston [2]** and **Allan Pring [1,2,*]**

1   South Australian Museum, North Terrace, Adelaide, SA 5000, Australia; neville.curtis@flinders.edu.au
2   College of Science and Engineering, Flinders University, Sturt Rd, Bedford Park, SA 5042, Australia;
    jason.gascooke@flinders.edu.au (J.R.G.); martin.johnston@flinders.edu.au (M.R.J.)
*   Correspondence: Allan.Pring@flinders.edu.au; Tel.: +61-8-8201-5570

**Abstract:** Our examination of over 230 worldwide opal samples shows that X-ray diffraction (XRD) remains the best primary method for delineation and classification of opal-A, opal-CT and opal-C, though we found that mid-range infra-red spectroscopy provides an acceptable alternative. Raman, infra-red and nuclear magnetic resonance spectroscopy may also provide additional information to assist in classification and provenance. The corpus of results indicated that the opal-CT group covers a range of structural states and will benefit from further multi-technique analysis. At the one end are the opal-CTs that provide a simple XRD pattern ("simple" opal-CT) that includes Ethiopian play-of-colour samples, which are not opal-A. At the other end of the range are those opal-CTs that give a complex XRD pattern ("complex" opal-CT). The majority of opal-CT samples fall at this end of the range, though some show play-of-colour. Raman spectra provide some correlation. Specimens from new opal finds were examined. Those from Ethiopia, Kazakhstan, Madagascar, Peru, Tanzania and Turkey all proved to be opal-CT. Of the three specimens examined from Indonesian localities, one proved to be opal-A, while a second sample and the play-of-colour opal from West Java was a "simple" Opal-CT. Evidence for two transitional types having characteristics of opal-A and opal-CT, and "simple" opal-CT and opal-C are presented.

**Keywords:** opal; hyalite; silica; X-ray diffraction; Raman; Infrared; $^{29}$Si nuclear magnetic resonance; SEM; provenance

## 1. Introduction

Opal is a generic term for a group of amorphous and paracrystalline silica species, containing up to 20% "water" as molecular $H_2O$ or silanol ($R_3SiOH$) or both [1,2]. Opals, both common opal and precious opal, have been the subject of considerable study over the last five decades or so [3]. Australia has long been the dominant sources of precious opal, exhibiting play-of-colour (POC) [4], but recently, opals from new fields, such as Ethiopia [5], Madagascar [6], Indonesia [7], Tanzania [8] and Turkey [9], have appeared on the market. This influx of new material on the market makes a re-examination of the classification opal timely.

Conventionally, opals are classified into three types [10]: opal-A (further divided into opal-AG (opal) and opal-AN (hyalite)), opal-CT and opal-C [11,12]. For nearly 50 years, the classification of opals proposed by Jones and Segnit [1], based on X-ray powder diffraction (XRD), has been widely adopted, e.g., [13–15]. The pertinent features of the XRD classification are:

- Opal-A (both AG and AN): broad absorption only, centred on 4.0 Å.
- Opal-CT: two prominent peaks at ~4.1 Å and 2.5 Å with a further peak showing variable degrees of separation at ~4.27 Å.

- Opal-C: prominent peaks at 4.04 Å and 2.5 Å.

The notation refers to "amorphous" (A) or is based on the similarity of the XRD reflection positions for $\alpha$-cristobalite (C) and $\alpha$-tridymite (T) [16,17]. The exact nature of atomic structures of opal-CT and opal-C remain largely unresolved. Opal-CT appears to be not just a fine-grained intergrowth of layers of cristobalite and tridymite, but a paracrystalline form of silica that has some structural characteristics of these minerals. Separate from the structural features revealed by XRD, some opals exhibit a play-of-colour. This is a textural feature related to the regular packing of silica spheres which are of a size to diffract visible light [4,10,12,15,18]. Both opal-A and opal-CT can show play-of-colour, but the atomic structure of the silica in the spheres is clearly different.

Other techniques such as Raman spectroscopy [13,19–21], $^{29}$Si nuclear magnetic resonance (NMR) [22–27], near infra-red spectroscopy [28–30] and neutron scattering [31] have also been used to try to unravel the complex structural relationships between the types of opal. Trace chemical analysis both of opal and associated minerals [7,15,18,32–36] may also provide evidence of provenance [37].

In this paper, we explore whether the XRD classification system is still valid for the newer-sourced material, particularly the precious or play-of-colour opals and that the terms opal-AG, opal-AN, opal-CT and opal-C represent homogenous structural groups. Transitions from opal-A to opal-CT to opals-C to quartz have been reported [38–43] and we take the opportunity to try to identify specimens showing evidence of intermediate forms. In addition to XRD, we will present results of techniques that focus on Si–O bonding to see if they provide evidence for homogeneity of the opal types. Techniques comprise Raman spectroscopy, far and medium infra-red (IR) spectroscopy, and single-pulse $^{29}$Si NMR. The key to this approach is the large suite of samples with all opals measured under similar conditions so that trends or differences readily come to light. We hope that this suite of opal samples will also be used by other researchers in their investigations.

The form of this paper is to focus on the results of the characterization of some 48 samples from new or unusual localities out of a total sample suite of some 230 samples that we have examined. A primary classification of all samples into groups according to the XRD methodology of Jones and Segnit [1] (opal-A, opal-CT and opal-C) was undertaken. Then selected samples were subjected to further study using Raman spectroscopy, infra-red spectroscopy and $^{29}$Si nuclear magnetic resonance spectroscopy.

## 2. Materials and Methods

Over 230 opal samples (opal-A, 67 samples; opal-CT, 161 samples; opal-C, 4 samples and 4 samples which appear to be intermediates between forms) were sourced from the South Australian Museum (G prefix), Flinders University (E), the Tate Collection (T) of the University of Adelaide, Museum Victoria (M), the Smithsonian National Museum of Natural History (NMNH) and through recent acquisitions (G NEW, OOC and SO). Where ambiguity exists, such as multiple samples in a single catalogued specimen lot, obvious differences between subsamples are indicated. About 250 individual specimens were analysed. We used well-documented specimens where possible and assumed that locality details were correct. A full list of the specimens examined is given in Supplementary Materials.

X-ray powder diffraction patterns (Bruker D8 Advance machine, Co source $K_\alpha$ = 1.78897 Å) were recorded with a scan speed of 0.0195° per second over the 2θ range 10 to 65°. Samples were ground under acetone before use. At least half of the samples were free from any obvious impurity. Literature [25,44,45] d-spacings of the major lines in the XRD patterns of the crystalline silica polymorphs were as follows: quartz: 4.25 Å and 3.33 Å, moganite: 4.43 Å, 4.38 Å, 3.39 Å (obscuring 3.33 Å), 3.10 Å and 2.86 Å, $\alpha$-cristobalite: 4.04 Å, 3.12 Å, 2.83 Å and 2.48 Å and $\alpha$-tridymite: 4.38 Å, 4.14 Å, 3.75 Å, 2.98 Å and 2.51 Å.

The major "impurity" was quartz, and this varied from minute traces to overwhelming amounts. A range of clays and related layer silicates was also noted. Only specimens with no or only trace amounts of impurities in the XRD pattern were included in the study.

Curve fitting for XRD data (d-spacing) gave peak positions, full-width half-maximum (FWHM) and relative proportions. A baseline spline was calculated using data read from the relatively flat

portions of the pattern and the Microsoft Excel Solver software was used to calculate the minimised least squares fit (about 1000 data points up to 45° 2θ). Since the major peaks spanned a wide range of 2θ, baselines showed variability (particularly for opal-A and some of the opal-CT samples) and only a generalised curve fitting regime was followed. The literature suggests a mix of Lorentzian and Gaussian types (pseudo-Voigt formalization) [46]. Fits could be obtained using either form or a mixture of forms, though this introduced extra variables but showed no obvious gain, and we adopted a pure Lorentzian peak shape for this initial study. Patterns were not corrected for $K_{\alpha 2}$. Consistent results from the large number of samples analysed for opal-A and opal-CT gave confidence in this approach.

Raman spectra were collected in the 100 to 1500 wavenumber ($cm^{-1}$) region, using a XplorRA Horiba Scientific Confocal Raman microscope. Spectra were acquired using a 50× objective (numerical aperture 0.6) at an excitation wavelength of 786 nm (27 mW measured at the sample) and spectrometer resolution of 4.5 $cm^{-1}$ FWHM. Typical integrations times for the spectra were 30 s and averaged from 6 (for chip samples) to 60 (for powdered samples). The instrument was calibrated using the 520.7 $cm^{-1}$ line of silicon and spectra were corrected to account for absorptions by the edge filter used to suppress the Rayleigh scattering peak. The spectrum of each sample was confirmed at several locations on the sample. Additional runs were made with laser wavelengths of 640 nm (3.8 mW) and 532 nm (7.3 mW) to confirm the Raman spectra recorded at 786 nm. Raman spectra of reference compounds are found in References [21,47–51]. There is, however, inconsistency in reported values for tridymite [52] with several types of Raman spectra despite having similar XRD patterns. Silanole [53] may also present a complicating Raman band at around 500 $cm^{-1}$. Several samples showed overwhelming fluorescence or gave a weak and largely featureless spectrum indicating that Raman is not as universal as XRD for characterisation. Because of the relatively low Raman scattering cross-section of opal, small amounts of "impurities" could produce misleadingly large peaks. All opals showed a significant baseline component that was most intense below 500 $cm^{-1}$. Baseline correction was not undertaken since the observed peaks were broad and the exact form of the baseline is unknown.

Attenuated total reflectance (ATR) infra-red (IR) spectra using a diamond ATR crystal were collected in two wavenumber ranges (100–600 $cm^{-1}$ and 400–4000 $cm^{-1}$). Spectra in the low wavenumber region were recorded at the Australian Synchrotron's THz-Far Infrared beamline using a Bruker IFS 125/HR spectrometer at a spectral resolution of 4 $cm^{-1}$. Four-hundred to five-hundred scans of the powdered samples were averaged to generate the final spectrum. For mid-IR, ATR spectra were collected using a Perkin–Elmer Frontier spectrometer and recorded at a 2 $cm^{-1}$ resolution. Overlapping wavenumber regions were examined and found to be comparable. The extended range provides full cover of the Si–O bonding derived peaks. Peaks consistent with water stretching (3000–3500 $cm^{-1}$) and bending (1600–1650 $cm^{-1}$) were noted but not further investigated for the current study. A simple intensity correction was applied by assuming the ATR penetration depth was directly proportional to the wavelength. We note that the spectra will give slightly different peak positions and band shapes to those obtained via transmission IR spectra or ATR spectra recorded with different crystals due to the anomalous dispersion (change of refractive index) across an absorption band.

The $^{29}$Si solid-state Magic Angle Spinning (MAS) NMR spectra were obtained using a Bruker Avance III 400 MHz spectrometer equipped with either a Bruker 4- or 7-mm probe with rotors spinning at 5 or 2.5 kHz, respectively. All spectra were collected at ambient temperature. Single-pulse (SP) experiments were typically carried out using a 90° pulse, high-power decoupling during acquisition (TPPM or SPINAL-64), followed by a recycle delay of 60 s. Spectra were referenced to 4,4-dimethyl-4-silapentane-1-sulfonic acid (DSS) at 0 ppm. It was noted that the opal-A samples required less sample or time to achieve good $^{29}$Si NMR spectra compared to other opal forms. The combination of the insensitivity of $^{29}$Si and the requirement for relatively large amounts of powdered sample (100 mg) limited our ability to measure numerous samples, thus only selected samples were used. The $Q_4$ ($RO_4Si$), $Q_3$ ($RO_3SiOH$) and $Q_2$ ($RO_2Si(OH)_2$) peaks were centred at around −112, −102 and −93 ppm [22] and overlapped owing to the broadness of the resonances (up to 10 ppm). The presence of a comparatively large $Q_3$ peak will affect the $Q_4$ peak position and, for this

reason, the spectra were deconvoluted. We have also measured [1]H and [29]Si [22] cross-polarisation (CP) MAS NMR spectra at various contact times allowing analysis of CP dynamics, and we will report these in a further paper (Curtis, Gascooke, Johnston and Pring, to be published). Reported chemical shifts ($Q_4$) [27,28] were: quartz −107.2 ppm to −107.1 ppm, cristobalite: −108.1 ppm and tridymite: −111.4 ppm and −109.3/−110.7/−114.0 ppm.

## 3. Results

The results are presented in two parts. The first is a summary of the results in terms of the overall classification of the opal family from the various techniques. The second presents the results for each opal group, focusing on the variation within the opal-CT group.

### 3.1. Overview

Figure 1 shows an overall appreciation of the XRD patterns, Raman, mid-IR spectra and NMR of the four groups of opal-AG, opal-AN (hyalite), opal-CT and opal-C using a set of exemplar specimens. As can be seen there are clear differences between opal-A, opal-CT and opal-C, though as will be discussed below the situation is more complex than this figure might suggest. Table 1 presents specimen data for the samples we have designated as exemplars (or typical examples) with a fuller list in the Supplementary Materials.

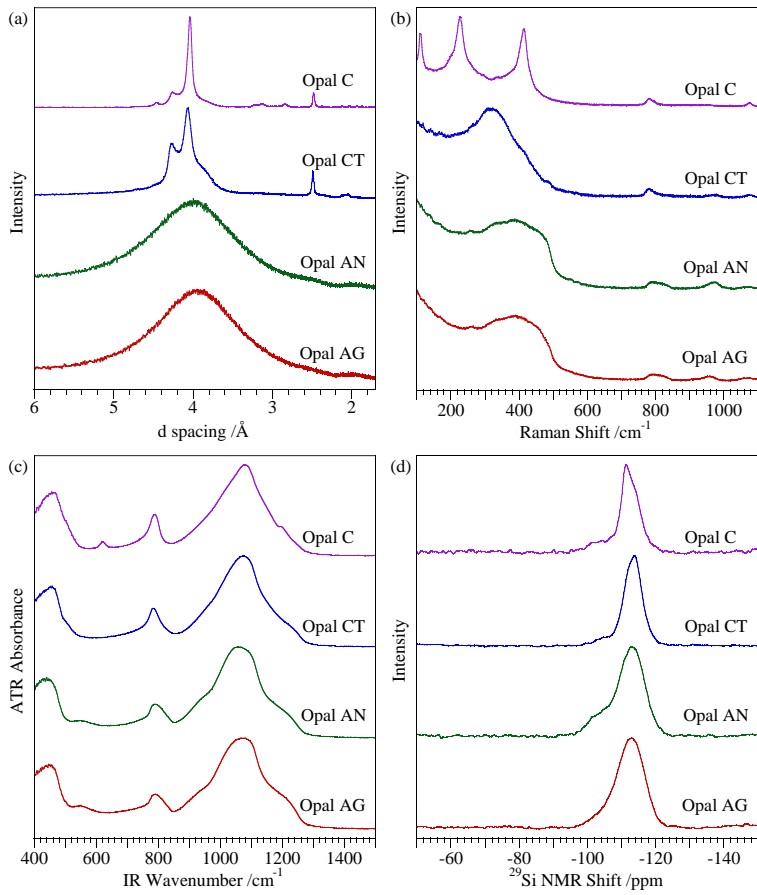

**Figure 1.** Patterns and spectra of typical samples showing (**a**) XRD patterns, (**b**) Raman spectra, (**c**) attenuated total reflectance mid-IR spectra and (**d**) single-pulse [29]Si MAS NMR spectra. In ascending order, the samples are: opal-AG (White Cliffs, Australia G13771) (red); opal-AN/hyalite (Valec, Czech Republic G32740) (green); opal-CT (Angaston, Australia, G9942) (blue) and opal-C (Iceland M5081) (purple). Spectra were scaled and offset for comparison.

**Table 1.** List of exemplar opal samples examined in detail in the study [a].

| Country | Location | Sample ID | Appearance | Type |
|---|---|---|---|---|
| Australia | White Cliffs, NSW | G1401 | Translucent no POC | A |
| Australia | Eurolowie, NSW | G1425 | Translucent pale brown glossy | CT |
| Australia | Iron Monarch, SA | G9620 | White glossy opaque | CT |
| Australia | White Cliffs, NSW | G8608 | White opaque POC | A |
| Australia | Unknown | G9260 | White to grey opaque | A |
| Australia | Two mile Coober Pedy, SA | G9594 | Translucent milky glossy minor POC | A |
| Australia | Four miles S of Angaston, SA | G9942 | Translucent white glossy | CT |
| Australia | Near Murwillumbah, NSW | G9964 | Slightly cloudy, clear POC | CT |
| Australia | Lightning Ridge, NSW | G13769 | Black, glassy band in matrix | A |
| Australia | White Cliffs, NSW | G13771 | Bag of samples | A |
| Australia | Angaston, SA | G24346 | Brown opaque | CT |
| Australia | Springsure, Qld | M8736 | Glassy (hyalite like) | A |
| Australia | Yinnar, Vic | T19006 | Glassy grey-brown | CT |
| Czech Rep | Valec, Bohemia | OOC11 | Glassy clear | A |
| Czech Rep | Valec, Bohemia | G32740 | Hyalite outgrowth, colourless | A |
| Ethiopia | Mezezo | G25374 | Deep-brown translucent | CT |
| Ethiopia | Afar | G32752 | Brown glass, some with POC | CT |
| Ethiopia | Mezezo | NMNH Eth1 | Pinkish POC on white | CT |
| Ethiopia | Yita Ridge, Menz-Gishe | G31892 | Nodules with clear orange centres | CT |
| Ethiopia | Mezezo | NMNH Eth 2 | Transparent brown POC | CT |
| Ethiopia | Wello | NMNH Eth 3 | Milky transparent POC | CT |
| Honduras | Unknown | G1441 | Milky transparent some POC | CT |
| Iceland | Unknown | M5081 | Opaque white | C |
| Indonesia | Cilayang Village, West Java | G34240 | Colourless with POC | CT |
| Indonesia | Mangarrai Prov, Flores | OOC6 | Translucent white | CT |
| Indonesia | Mamuju, West Sulawesi | OOC13 | Blue-green matrix of "grape agate" | A |
| Kazakhstan | Voznesenovka, Martuk | M53407 | Orange glass | CT |
| Kazakhstan | Zelinograd | G32925 | Translucent vermilion glassy | CT |
| Madagascar | Bemi, Befotaka District | G NEW05 | Clear yellow | CT |
| Madagascar | Bemi, Befotaka District | G NEW07 | Translucent pale brown | CT |
| Mexico | La Trinidad Queretaro | G31851 | Single piece with opal inclusions | CT |
| Namibia | Khorixas district | G NEW29 | Blue to white opaque | CT |
| Peru | Acari | G33912 | Massive blue | CT |
| Spain | Mazarron, Murcia | OOC4 | Composite with green zones | CT |
| Tanzania | Kigoma, Region | G NEW19 | Pale orange shades glassy | CT |
| Tanzania | Haneti | G NEW03 | Opaque green | CT |
| Tanzania | Haneti | G NEW04 | Opaque green, some glassy zones | CT |
| Tanzania | Arusha | G34238 | Transparent green layer | CT |
| Turkey | Kutahya | G NEW24 | Translucent green and brown | CT |
| Turkey | Eskisehir | G NEW25 | Opaque white with indigo speckles | CT |
| Turkey | Anatolia | G NEW26 | Opaque white transparent green inside | CT |
| Turkey | Yozgat, Anatolia | G NEW27 | Blue-green transparent glass | CT |
| Turkey | Yozgat, Anatolia | G NEW28 | Olive-green transparent glass | CT |
| USA | Opal Butte Mine, Oregon | G NEW18 | Glassy white | CT-C |
| USA | Manzano Mtns. New Mexico | G NEW30 | White opaque mass | CT |
| USA | Virgin Valley, Nevada | G31852 | Milky and translucent zones | CT |
| USA | Virgin Valley, Nevada | G32263 | Translucent brown | CT |
| USA | Virgin Valley, Nevada | M19717 | Opaque glassy POC | CT |
| USA | Virgin Valley, Nevada | OOC5 | White and POC zones | CT |

[a] Some samples yielded more than one experimental sample.

The form of the XRD patterns shown in Figure 1a matches those given in the original opal classification by Jones and Segnit [1] published in 1971. We found that for the most part samples taken from the same specimen, though differing in colour or texture, showed similar, but not necessarily identical, XRD patterns. We note that the difference between opal-CT and opal-C may be subtle

when faced with a single sample, as assignment to one group or the other can be difficult without detailed comparison.

The major absorption for the Raman spectra for exemplar specimens (Figure 1b) was in the 200–500 $cm^{-1}$ region with isolated smaller peaks up to 1100 $cm^{-1}$, and the spectra for opal-A, opal-CT and opal-C were distinct. Spectra were consistent between the unground and the powdered samples (which was also used in XRD). Generally, samples were mostly homogenous but, in some cases, extra peaks were observed in the spectra. These were probably non-opal impurities such as inclusions of silicate minerals. The most likely "impurity", quartz, has a characteristic spectrum of a relatively strong, isolated, sharp peak at 459 $cm^{-1}$ and lesser ones at 108 $cm^{-1}$ (sharp) and 227 $cm^{-1}$ (broad).

All ATR-IR spectra (Figure 1c) for opals show a common set of peaks at around 470, 790 and 1080 $cm^{-1}$. Below 400 $cm^{-1}$, peaks were either non-existent or very weak, and thus measurement of spectra in the 400-1600 $cm^{-1}$ range is needed to provide unambiguous differentiation of opal-A, opal-CT and opal-C. The IR spectra of reference compounds are found in References [48–50,54]. Quartz may be identified by peaks at 697 and 780 $cm^{-1}$ which are at lower energy to the common peak at 795 $cm^{-1}$, seen for both quartz and all the opal samples. Both opal-A and opal-C showed distinct peaks that were absent for opal-CT (see later). A complication occurred with the far-IR as a range of spectra were seen (Figure 2a). The peak at around 470 $cm^{-1}$ showed a progressive trend of broadening, shifting to lower wavenumber and the appearance of a second peak at around 440 $cm^{-1}$. This was found to be common for all types of opal. There was no obvious correlation with the other spectral methods or with the behaviour in the mid-IR. Figure 2b also shows variation in the mid-IR range (in this case for opal-A for which it is most pronounced).

The $^{29}Si$ MAS NMR spectra are potentially discriminating for the opal types, although the average $Q_4$ peak positions were close for all the forms. Some difference was seen for FWHM and the positions of the $Q_3$ peaks. Peaks may be readily curve-fitted to differentiate the $Q_4$ and $Q_3$ peaks (Figure 3).

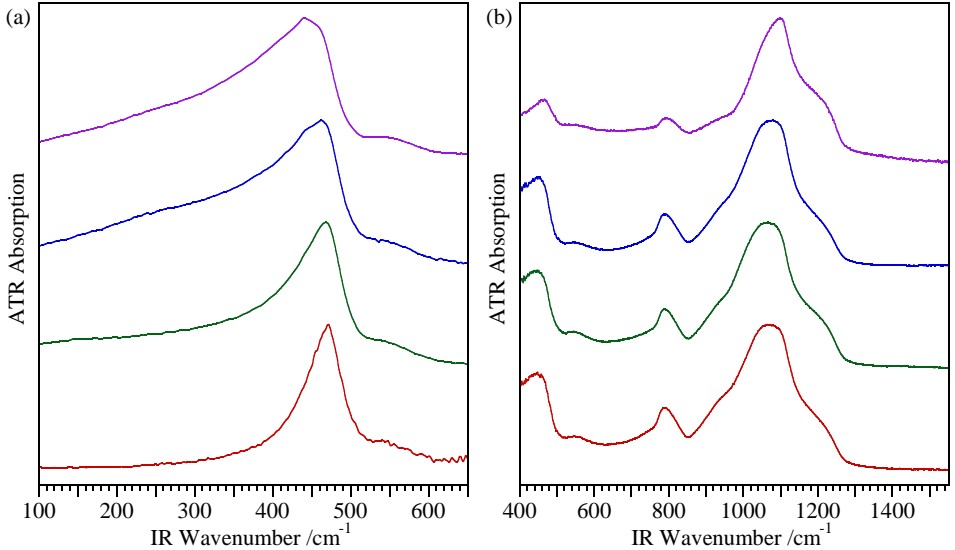

**Figure 2.** Far IR spectra of opal-AG samples showing the range of types. In ascending order: (**a**) White Cliffs, Australia (G8608), Lightening Ridge, Australia (G13769), Coober Pedy Australia (G9594) and Iron Monarch, Australia (G9260). (**b**) Mid-IR spectra of opal-A samples showing the range of types. In ascending order: Valec, Czech Republic (OOC11) (opal-AN), Coober Pedy Australia (G9594) (opal-AG), Springsure, Australia (M8736) (opal-AN) and White Cliffs, Australia (G1401) (opal-AG). Spectra were scaled and offset (*y*-axis) for comparison.

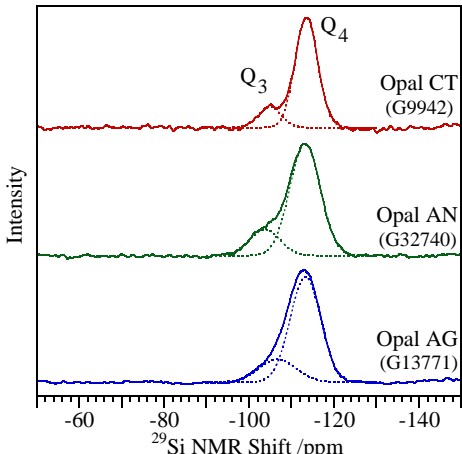

**Figure 3.** Experimental and fitted curves for $^{29}$Si MAS NMR spectra. The exemplar opal-AG, opal-AN and opal-CT samples: opal-AG (White Cliffs, Australia G13771) (red), opal-AN/hyalite (Valec, Czech Republic G32740) (green) and opal-CT (Angaston, Australia, G9942) (blue).

*3.2. Opal-A*

All samples showed broad and weak XRD patterns with absorption centred on ~4.0 Å (Figure 1a) and had a poorly defined baseline compared to the other opal types. There was also a weak broad band at around 2.0 Å. Consistency was demonstrated through curve fitting with peak maxima of 3.98 ± 0.04 Å and FWHM 0.53 ± 0.02 Å. There was little, if any, difference between the opal-AG and opal-AN samples.

Opal-A Raman spectra were distinct from the spectra for opal-C and opal-CT in the 100–600 cm$^{-1}$ range, so this provided a secondary delineation of the opal type. There were no obvious differences between Raman spectra for opal-AG and opal-AN in the 100–600 cm$^{-1}$ range. Raman spectra between 700 and 1200 cm$^{-1}$ (Figure 4) showed several weak peaks and a broad peak (probably two peaks) in the range 760–860 cm$^{-1}$ that was characteristic of both opal-AG and opal-AN and which also separated these from most examples of opal-CT and opal-C. In general, the middle peak in Figure 4 was at 960–965 cm$^{-1}$ for opal-AG and 970–975 cm$^{-1}$ for opal-AN, though the peaks were weak. The IR spectra (Figure 1c) for opal-A were characteristic and distinct to those for opal-CT and opal-C with the peak at around 550 cm$^{-1}$ being specific for opal-A, but there were no obvious differences between opal-AG and opal-AN.

The $^{29}$Si MAS NMR spectra (Figure 1d) were potentially discriminating for opal-A, as the FWHMs tended to be larger for it than for other types. The $Q_4$ peak positions were −113.3 ± 0.2 ppm with FWHM of 8.5 ± 0.2 ppm for opal-AG and −113.4 ± 0.3 ppm and 8.7 ± 0.6 ppm for opal-AN. These FWHMs were higher than the 6.5 ± 0.6 ppm seen for opal-CT. All samples showed significant amounts of $Q_3$ peaks within the range 10–40%. The visual difference of opal-AG and opal-AN in Figure 1d can be traced to the placement of the $Q_3$ peaks with the former at −106.5 ± 0.9 ppm and the latter at −103.8 ± 0.8 ppm (see Figure 3).

Of the more recent finds, the massive blue-green opal that occurred as the base of some of the purple "grape agate" from near Mamuju, West Sulawesi, Indonesia was an opal-AG. A previous report [37] suggested that this material might have been a clay.

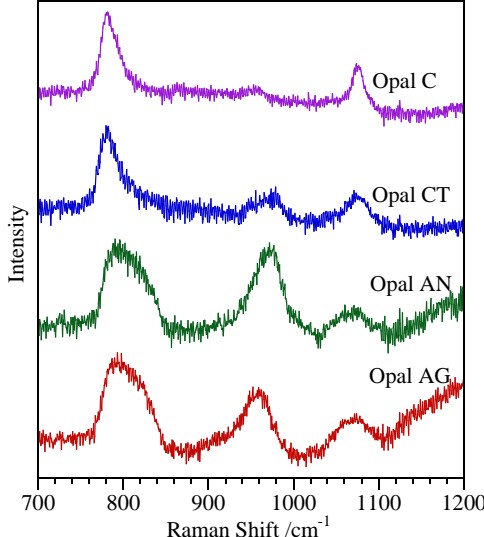

**Figure 4.** Raman spectra (700 to 1200 cm$^{-1}$) of opal samples. In ascending order: White Cliffs, Australia (G13771) (opal-AG), Valec, Czech Republic (G32740) (opal-AN), Angaston, Australia (G9942) (opal-CT) and Iceland (M5081) (opal-C). Spectra were scaled and offset for comparison.

### 3.3. Opal-CT

All samples showing substantial XRD peaks at ~4.1 Å and 2.5 Å may be considered as opal-CT according to the Jones and Segnit [1] classification. However, we believe that this is an oversimplification. The maximum was distinct from that for the opal-C grouping (~4.1 Å to 4.04 Å, respectively), though this was only apparent when two samples were directly compared. The increasing complexity of the main peak ~4.1 Å in the XRD pattern (Figure 5), presented in terms of increasing sharpness of the peak at 2.5 Å) suggests that the term opal-CT does not represent a homogenous group. We noted a trend whereby a peak at 4.28 Å was sometimes absent, sometimes present as a shoulder and sometimes as a separate peak. This was noted in a number of previous studies [55–57]. Complexity of the peak at ~4.1 Å appears associated with the peak width at 2.5Å. It is also apparent from the large number of samples measured here (161 specimens) that the patterns varied according to the relative peak heights and widths in the composite at around ~4.1 Å (see later). Sample G32752 (Afar, Ethiopia) (Figure 5) showed a simple, near symmetric XRD peak at 4.1 Å, as well a broad peak at 2.5 Å. At the other extreme, sample G NEW19 (Kigoma, Tanzania) showed two sharp peaks and a prominent shoulder in the composite at around ~4.1 Å and a sharp peak at 2.5 Å.

We note that the more "simple" opal-CT specimens include POC specimens from Ethiopia (G25374, G31892, G32752 (three distinct samples from the same specimen), NMNH Ethiopia samples 1, 2 and 3), Honduras (G1441), the USA (M19717, OOC5), Mexico (NMNH 117414) and Australia (G9964). The group also includes mostly transparent samples from worldwide localities such as Turkey (G NEW24, G NEW26, G NEW27, G NEW28), Madagascar (G NEW05), Mexico (G34738, NMNH115816, NMNHR1694), Australia (T19006, T23363, M12495, M20970), Peru (G33912), Brazil (T1152), Indonesia (OOC6), Spain (G NEW12) and the USA (G9116, G31852, G34243). The converse is not, however, true, with for instance G NEW07 from the same site in Madagascar as G NEW05 showing a more complex pattern. Some samples showing POC also have more complex XRD patterns.

We also noted a distinct change in the shape of the background of the XRD pattern. In the simpler opal-CT patterns, the background was distinctly lower on the low-angle side and higher on the high-angle side of the peak at 4.1 Å. In contrast, the background around this peak became more uniform as the peak at 4.1 Å became more complex.

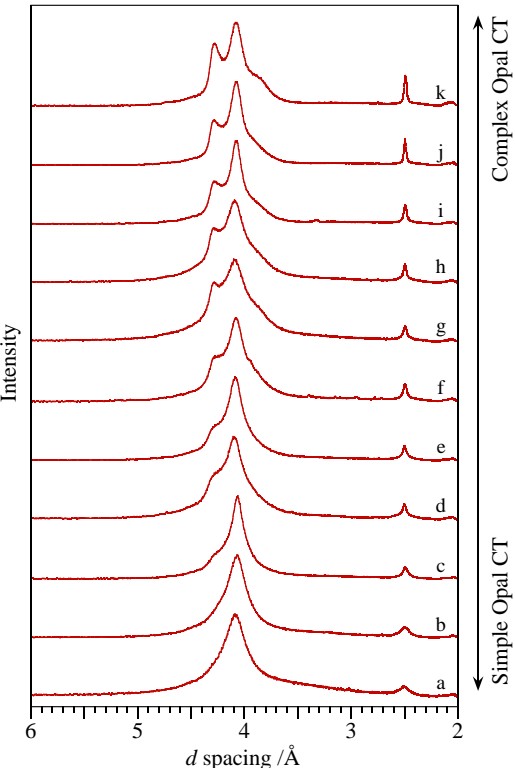

**Figure 5.** A series of XRD patterns illustrating the structural changes across the opal-CT group. The patterns were arranged in order of increasing complexity of the main peak at ~4.1 Å. Any subdivision between "simple" opal-CT and "complex" opal-CT is arbitrary, but patterns *a* and *b* are clearly distinct from patterns *i*, *j* and *k*. Note the progressive change in the sharpness and shape of the reflections at 4.1 and 2.5 Å. The specimens are (in ascending order from simplest to most complex): (**a**) Afar, Ethiopia (G32752), (**b**) Mezezo, Ethiopia (NMNH Eth 1), (**c**) Murwillumbah, Australia (G9964), (**d**) Acari, Peru (G33912), (**e**) Honduras (G1441), (**f**) Indonesia (OOC6), (**g**) Kazakhstan (M53407), (**h**) Nevada, USA (G32263), (**i**) Tanzania (G34238), (**j**) Nevada, USA (G31851) and (**k**) Tanzania (G NEW19).

We explored peak fitting to investigate the heterogeneity seen in opal-CT to see if quantifiable support may be gained for the continuum of "simple" and "complex" opal-CT. As Figure 6 shows, a reasonable proposition is that there were three peaks present (P1 to P3) in the 4.1 Å complex with a further one at 2.5 Å (P4). Well-fitting deconvolutions were obtained for all relatively pure (i.e., no visible or only a very minor amounts of quartz or other impurities) samples and confirms three peaks with positions at 4.27 ± 0.01 Å (P1), 4.08 ± 0.01 Å (P2) and 3.89 ± 0.03 Å (P3). The peak at 2.50 ± 0.01 Å (P4) was also constant.

The peak areas were normalized to P2 and the FWHM examined for each peak used to explore trends. A strong correlation in FWHM was for P1 and P4 (Figure 7a) where the linear trend between "simple" and "complex" extremes is manifest. The "simple" opal-CTs plot in the top right-hand corner are Ethiopian opals with POC. Other POC opals are distributed though the plot suggesting that this textural effect is independent of crystal–chemical features. It is also worth noting that other textural features such as the alignment and size of spheres are also implicated in POC.

There is also a linear correlation for the relative areas of P1 and P3 peaks as shown in Figure 7b, implying that, in simplistic terms, that this could be considered to represent the increase in the "tridymite" component comparted to the "cristobalite" (P2 coinciding with the overlap of the C and T contribution). The presence of trace amounts of quartz does not seem to affect this trend, but the problem associated with fitting caused by the asymmetric baseline may affect the ratio and over-represent P1 or P3.

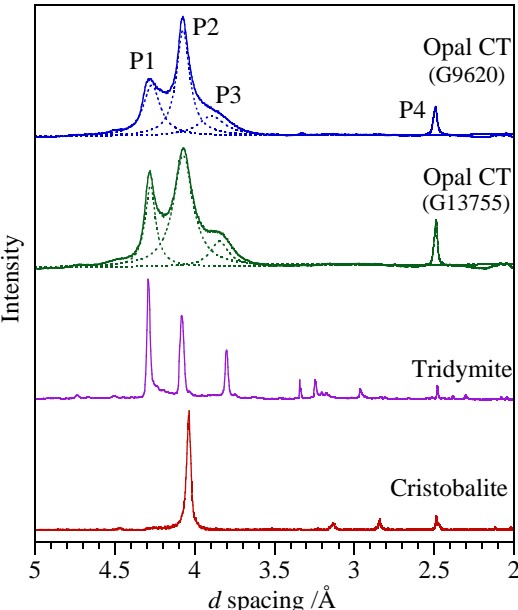

**Figure 6.** Comparison of mixed XRD patterns for mineral samples of tridymite (G1395) and cristobalite (RRUFF Database ID R060648) with Ross, Tasmania, Australia (G13755) and Iron Monarch, Australia (G9620) showing curve-fitting elements and actual pattern. Spectra were scaled and offset for comparison.

The implication of the separate relationships shown in Figure 7 is that the XRD pattern for any particular opal-CT will be hard to predict. Peaks may be large or small and sharp or narrow. We think it unlikely that a single parameter may be derived to quantify an opal-CT. Principal component analysis may provide a solution, but this would need to have a chemical or structural basis rather than mathematical manipulation. We noted that the two points at the top right of Figure 7b (G9960 from Iron Monarch, Australia and G NEW17 from Turkey) lie in the middle of the plot in Figure 7a. Similarly, G13755 (Ross, Tasmania, Australia, bottom left of Figure 7a) which shows very prominent and sharp XRD peaks (see Figure 6) lies in the centre of Figure 7b. It is also unlikely that the XRD pattern will allow prediction for the potential of POC from that locality.

The Raman spectra Figure 8 also demonstrated that opal-CT is not a homogenous group as shown by the spectra in Figure 5. An arbitrary classification may be made according to the rationales: (i) featureless spectrum with a maximum above 300 cm$^{-1}$, (ii) signs of structure with maximum at or below 300 cm$^{-1}$ and (iii) more developed spectrum with partially delineated peaks. It is not clear if the trend was due to the change in structure with different spectra or merely a sharpening of existing peaks. We noted that those samples with few features in the Raman were mostly coincident with the "simple" opal-CT points at the upper right of Figure 7b (see Figure 8b). The most Raman-structured samples tended to lie at the lower left of Figure 8b and thus correlated with the more complex XRD patterns.

The 700 cm$^{-1}$ to 1000 cm$^{-1}$ Raman region showed potential differences between the "simple" and "complex" opal-CT examples in the 850–1000 cm$^{-1}$ region. Unfortunately, the spectra were generally weak and were indicative rather than definitive. This is further complicated as not all samples were amenable to Raman due to the fluorescence, and this limits the applicability of the technique.

The $^{29}$Si MAS NMR data showed that SP opal-CT $Q_4$ peak positions were marginally downfield of those for opal-A at −113.7 ± 0.3 ppm with FWHM 6.5 ± 0.6 ppm. In general, the $Q_3$ peaks were visible and were at −105.3 ± 0.8 ppm. The FWHMs of 6.5 ± 0.6 ppm were smaller than for opal-A at 8.4 ± 0.4 ppm. We also found that some of the "simple" opal-CT types gave visually different spectra that could not be deconvoluted satisfactorily.

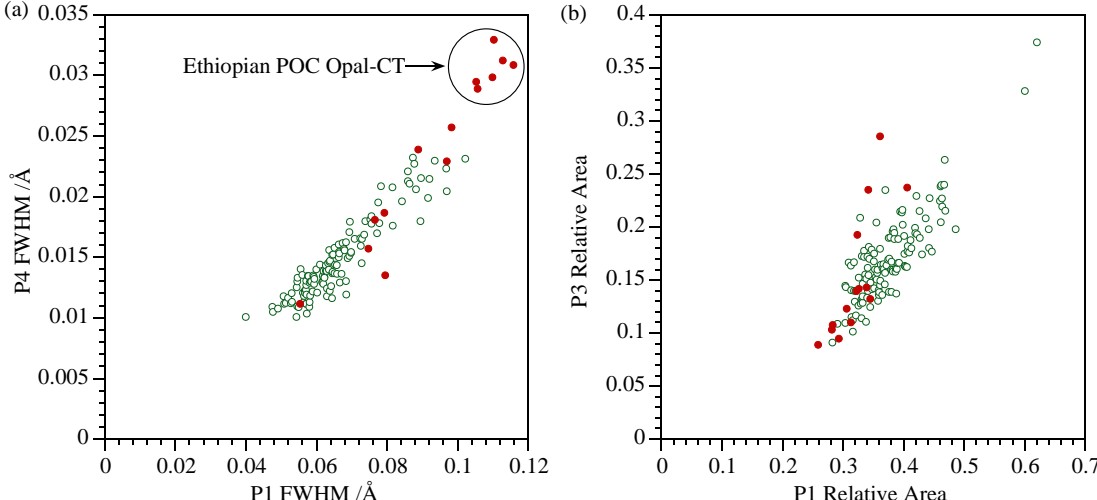

**Figure 7.** Correlations of XRD curve fitting data: (**a**) FWHM for P1 (*x*-axis) and P4 (*y*-axis), and (**b**) relative amounts of the P1 (*x*-axis) and P3 (*y*-axis) peaks (P2 is set at unity). Samples showing play-of-colour (POC) are shown as red-filled circles, whereas non-POC samples are represented by green-open circles. The subset of samples from Ethiopia displaying POC are circled in panel (**a**).

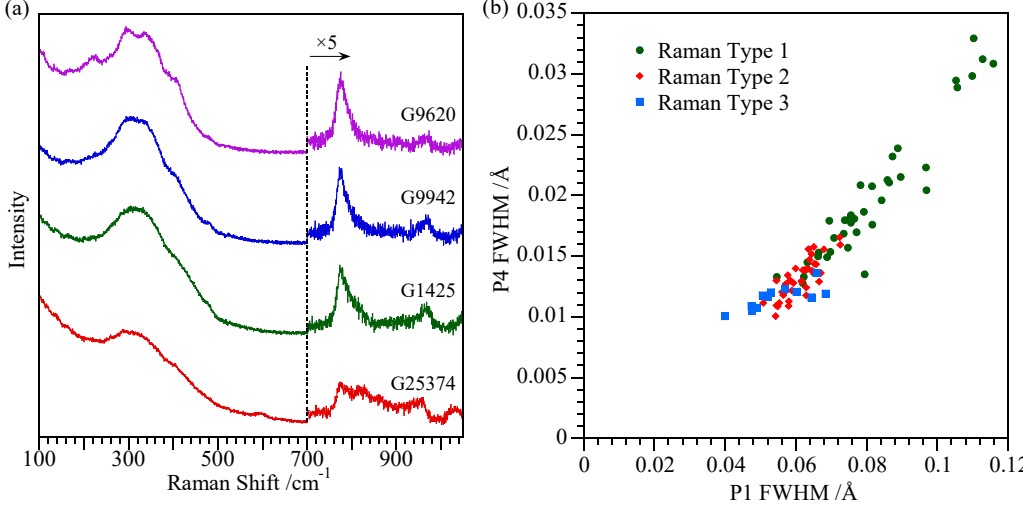

**Figure 8.** Raman spectra of opal-CT showing progressive structure. In ascending order: (**a**) "simple" opal-CT from Mezezo Ethiopia (G25374), and increasingly complex forms from Eurolowie, Australia (G1425), Angaston, Australia (G9942) and Iron Monarch Australia (G9620). (**b**) Plot of the XRD pattern FWHM of P4 versus P1 separated into different Raman types. See text for details regarding the definitions of the Raman types observed in this study. Not all samples yielded a Raman spectrum due to the problems with fluorescence.

### 3.4. Opal-C

Only four discrete samples were identified as opal-C in this study. Two of these had zones of both transparency and opaqueness though having identical XRD patterns. All showed very small peaks for quartz. The common feature was a large peak at 4.04 Å with a smaller one at 4.28 Å. The small pair of equally sized peaks at 3.11 Å and 2.84 Å was diagnostic of cristobalite [44].

Raman spectra (Figure 1b) showed characteristic spectra similar to that of cristobalite [51] at low wavenumber (peaks at 107, 222 and 409 cm$^{-1}$), while the medium wavenumber spectrum shown in Figure 4 is possibly diagnostic but again weak. The combined far- and mid-IR for the opal-C samples showed these features: 300 cm$^{-1}$ (sh), 385 cm$^{-1}$ (sh), 480 cm$^{-1}$ (m), 625 cm$^{-1}$ (w), 795 cm$^{-1}$

(m), 1090 cm$^{-1}$ (s) and a shoulder at 1200 cm$^{-1}$. The most delineating feature was the band at around 625 cm$^{-1}$ which was visible both via far- and mid-IR. We did not, however, find any evidence of an IR band at 145 cm$^{-1}$ as was reported in a previous study [50].

The $^{29}$Si MAS NMR spectra of the opal-C samples showed variability and the Iceland sample (M5081) shown in Figure 1d should be treated as an example rather than a typical spectrum. This had characteristics of a peak position of −114.4 ppm and an FWHM of 5.7 ppm. Spectra suggest more complexity than those seen for opal-A and opal-CT with the likelihood of additional peaks.

### 3.5. Transitional Samples

### 3.5.1. Samples Showing Opal-A and Opal-CT Characteristics

Three samples had XRD patterns which had characteristics of both opal-A and the simpler form of opal-CT, with a very broad peak at 4.1 Å and a relatively small and broad peak at 2.5 Å. Sample OOC4 from Mazarron, Spain had a major peak FWHM estimate of about 6.3°, while that for T22824, from Megyasro, Hungary, was 4.3°. Neither gave a satisfactory XRD curve fitting. Not shown is E1950 (Canungra Mts. Australia) which had an FWHM of around 3° and which could be XRD curve fitted (and plottted as a "simple" opal-CT). This contrasts with a value of about 8° for a typical opal-A and less than 2° for the POC opal-CT from Mezezo, Ethiopia (G25374) and the remainder of the simple opal-CT samples. Only T22824 yielded a Raman spectrum and this was consistent at both low and medium wavenumbers with opal-A. The orange opal-CT from Voznesenovka, Kazakhstan (T22824) and OOC4 from Mazarron, Spain showed a peak at 550 cm$^{-1}$ in the IR though this was not readily apparent in E1950 (Figure 9).

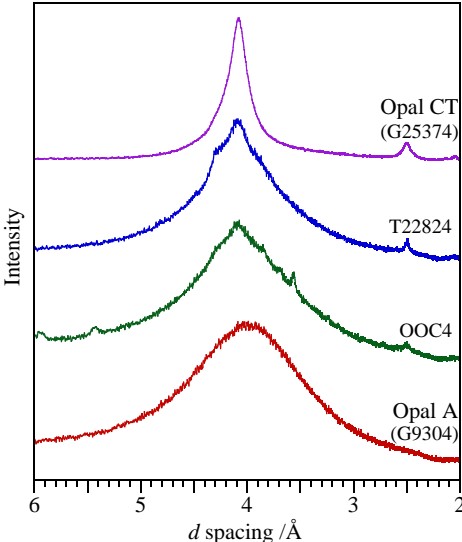

**Figure 9.** Transitional opal XRD pattern (lower-middle OOC4 from Mazarron, Murcia, Spain, upper-middle T22824 from Megyasro, Hungary). Shown with G9304 (opal-A, lower) and G25374 (simple opal-CT, upper). Scaled and offset (*y*-axis) for comparison.

Raman spectra showed broad peaks at 760–860 cm$^{-1}$ (as in opal-A) and 975 cm$^{-1}$ (as in hyalite). While there were coincidences in position, it is difficult to assert that the XRD peak at 2.5 Å represents a narrowing of the broad and weak second peak in opal-A. Critically, the XRD patterns and Raman spectra showed no evidence of cristobalite, so a reasonable assumption was that these represent transitional opal-A/opal-CT. The SEM images (Figure 10) were not consistent with opal-AG.

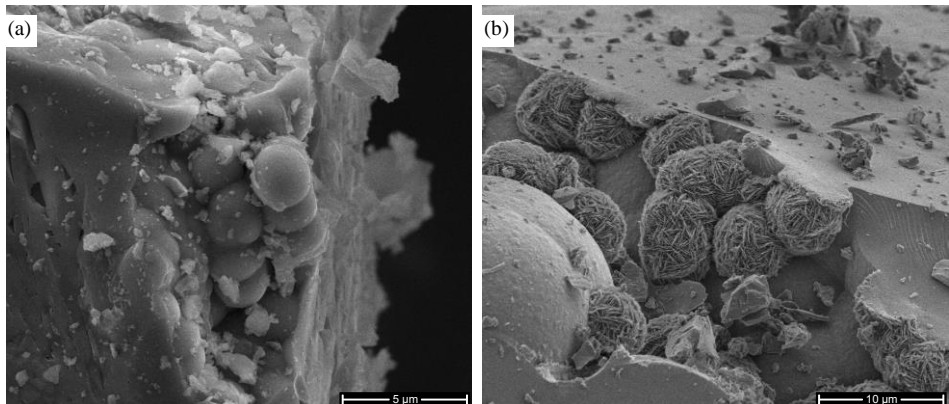

**Figure 10.** SEM of: (**a**) OOC4 AND (**b**) T22824 (RHS) showing large spheres and bundles of plates.

### 3.5.2. Samples Showing Opal-CT and Opal-C Characteristics

The sample from the Opal Butte Mine, Oregon USA (G NEW18) appears to be a transitional form of opal-CT to opal-C as the characteristic cristobalite peaks are present in both the XRD pattern and Raman spectrum (Figure 11). The XRD maximum is at 4.02 Å which is consistent with the cristobalite-like patterns of opal-C as are the two small peaks between 3.14 Å and 2.85 Å. The form and position of the Raman absorption at low wavenumbers are not characteristic of opal-A and are more like that seen for the "simple" opal-CT samples with an overlay of cristobalite peaks. At medium wavenumbers only one peak is seen at around 800 cm$^{-1}$ while the peak at just below 1000 cm$^{-1}$ is perhaps more significant than is seen for the opal-C samples. A weak response at 625 cm$^{-1}$ in the IR is also consistent with opal-C as are the patterns at lower wavenumber. Overall this suggests a transitional "simple" opal-CT to opal-C species.

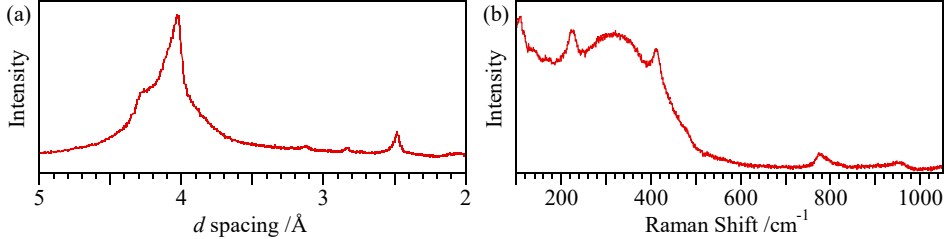

**Figure 11.** Transitional opal CT to C from Opal Butte Mine, Oregon USA (G NEW18). (**a**) XRD pattern and (**b**) Raman spectrum.

## 4. Discussion

### 4.1. Applicability of XRD for Primary Classification

We found that all opal samples, including those from "newer" sources, including Ethiopia, Indonesia, Kazakhstan, Madagascar, Peru and Tanzania could be readily classified using XRD into one of the Jones and Segnit [1] groups. All of the opals for newer localities were opal-CT, except for two of the Indonesia samples which were Opal-A (Table 1). XRD provides a ready and informative delineation of opal-A, opal-CT and opal-C but the identification of transitional forms requires additional data from spectroscopic techniques. Differentiation of opal-AG and opal-AN is, however, not possible from XRD, but can usually be readily seen visually. The two types of precious opal (opal-A and "simple" opal-CT) can be readily separated by XRD.

Further detail may be gleaned, particularly of opal-CT, through semi-quantitative curve fitting.

The work described here provides an alternative analysis to previous studies [14,17,55,58] where the maximum and width of the composite peak at 4.1 Å were interpreted in terms of contributions from cristobalite-like and tridymite-like components.

*4.2. Homogeneity and Characterisation of Opal Groups*

4.2.1. Opal-A

This group was clearly identified by the distinctive broad XRD pattern, the featureless but broad Raman pattern centred at 370 cm$^{-1}$ (uncorrected) and a specific IR peak at around 530 cm$^{-1}$. The $^{29}$Si NMR will show a relatively broad (8.5 ppm) and symmetric $Q_4$ peak at around −113.3 ppm hiding the $Q_3$ component. Hyalites (opal-AN) were visually different from the opal-AG group being botryoidal and gel-like though with similar XRD patterns and Raman and IR. The major $^{29}$Si NMR spectra will show resolved peaks for $Q_4$ and $Q_3$ unlike for opal-AG.

4.2.2. Opal-CT

The XRD patterns for this group are characterized by peaks at 4.1 Å and 2.5 Å. With the 4.1 Å reflection range from a "simple" asymmetric peak accompanied by broad weak reflections at 2.5 Å, to a "complex" broad group of three peaks centred on 4.1 Å and a sharp reflection at 2.5 Å. The opal-CT that gave the simple X-ray patterns were more or less transparent, though they may have been coloured and showed POC to some extent, while those with complex patterns tended to be opaque and included material that was generally considered common opal, though there were examples with POC. The Raman spectra for those that gave "simple" XRD patterns were generally weak and featureless showing only a peak around 300 cm$^{-1}$, while those for the opal-CTs with more complex XRD patterns (if obtainable) showed broad absorption in the 200 to 500 cm$^{-1}$ range and varying degrees of structure, possibly with discrete peaks at 220, 295, 340 and 410 cm$^{-1}$. The IR spectra and $^{29}$Si MAS NMR across the range of opal-CTs did not provide differentiation.

These "simple" opal-CTs included the Ethiopian play-of-colour opals as well as some Australian specimens, but were not from the current active opal fields. For instance, G9964 is labelled as "jelly opal" from Murwillumbah in New South Wales and has a cloudy but transparent appearance. It has been proposed that Australian POC opal is sedimentary if opal-A and volcanic if opal-CT [59]. Other sources for "simple" opal-CT without play-of-colour include: Brazil, France, Honduras, Indonesia, Madagascar, Mexico, Peru, USA (Nevada). Published spectra show further examples of these simple XRD patterns, e.g., [7,13,19,60].

Recent measurements suggest that POC in opal-CT is also caused by the presence of diffracting patterns of spheres [5,61] similar in concept to those for opal-A [4]. Our work on the chemical characteristics of opal-CT, however, can provide no light for prediction of POC, though we note that an Ethiopian opal with a wide peak at 2.5Å is likely to show the effect. Other POC examples do not show any correlation with XRD patterns or spectroscopic measurements.

4.2.3. Opal-C

There is no visible difference between these and opal-CT samples. Some show the feature of transparent specimens merged with white translucent areas [49] which have the same XRD patterns, Raman and IR spectra. The distinct features are XRD pattern peaks at 4.04 Å and 2.5 Å coupled with a pair of small peaks between the major ones. Raman shows a characteristic pattern with peaks 107, 222 and 409 cm$^{-1}$. The IR spectra were distinct with a small but clearly defined peak at 625 cm$^{-1}$. The $^{29}$Si MAS NMR showed 6 ppm FWHM peak at −113 ppm.

*4.3. Spectroscopic Characterisation Techniques*

Raman spectroscopy [6,7,13,20,32] can be used in non-destructive mode with data gained in the far- and mid-range providing information related to Si–O bonding, although direct assignment of

bands to the opal structure is difficult. The spectra for opal-A, opal-CT and opal-C were distinct and provided ready differentiation. With some care, hyalite may also be identified if it is not already apparent from its distinctive appearance. Raman spectroscopy is not as viable a technique as XRD for two major reasons [62]. Firstly, opal has a relatively low cross-section for Raman scattering resulting in weak signals, thus requiring long collection times. Second, many samples exhibited a large degree of fluorescence which swamps the signal. The "complex" opal-CT samples were more likely to be fluorescent than the "simple" types, indicating possibly higher content of metal impurities in "complex" opal-CT. Opal-C samples, and some proposed transitional forms, had similar peaks to that of $\alpha$-cristobalite and this is probably the preferred means for identification. The current work is consistent with previous studies.

ATR IR has been used previously to examine silica species [1,54] but has been little exploited recently for opal. It does, however, represent an alternative to XRD as opal-A, opal-CT and opal-C can be readily discriminated, though without the additional information relating to "simple" or "complex" forms. While we used ground samples, it could be a non-destructive method in ATR mode. Although not performed here, we suggest that reflectance IR spectroscopy may also be a valuable tool for non-destructive examination of opal samples. The $^{29}$Si NMR is of more interest for investigation of the chemical structure of opal rather than as a differentiation technique.

### 4.4. Comments on Nature of Opal-CT

This has been the subject of controversy for some years. The term opal-CT can be interpreted as zones of cristobalite and tridymite or as an intimate intergrowth, whether regular or disordered. In our opinion, we feel that the term "opal-CT" is a misnomer. The notation was based on a similarity of XRD peak positions of $\alpha$-cristobalite and $\alpha$-tridymite with those in opal samples [1,25,44]. This has been complemented by modelling studies of XRD patterns [63] and Raman spectra [52,64]. The XRD patterns have been analysed in terms of $\alpha$-cristobalite to $\alpha$-tridymite ratio [14]. The TEM images have been interpreted in terms of the intergrowth of domains of cristobalite and tridymite and as tridymite-like stacking faults in cristobalite [55,56]. As Figure 6 implies, a reasonable proposal based on peak positions of reference compounds is that the opal-CT peaks derive from cristobalite and tridymite, with P1 due to tridymite, P2 due to cristobalite and tridymite and P3 due to tridymite (and possibly cristobalite). While the XRD curve fitting suggests complexity in the structure, it is not clear how many discrete species may be involved, as correlation was noted between the intensities of the P1 and P3 peaks in one instance and the sharpness of the P1 and P4 peaks in another. The trending evidence (Figure 7) did not imply this. If tridymite was present, then we might expect the P1, P2 and P3 peaks to be linked in some way. P3 did not correlate in sharpness with P1 and P4 and was shifted compared to $\alpha$-tridymite. We also believe that the Raman evidence was equivocal, as while we find (baseline uncorrected) peaks at 220, 295, 340 and 410 cm$^{-1}$ in the most structured opal-CT samples, we do not believe that the spectra were of sufficient quality to add anything significant to the issue.

The notion that opal-CT is a disordered, intimate mix of cristobalite and tridymite has also been questioned on the basis on XRD and Raman data [52,57,64,65]. The lack of evidence for the presence of cristobalite led to the proposal for "opal-T" based on interpretation of the Raman data [52,57,65]. Whether this is a "not cristobalite" rather than a "positive tridymite" assignment is a moot point. Supporting evidence for tridymite, however, presents a major problem with the potential multitude of stacking variations of this structure [44]. We found no evidence for the triplet reported for tridymite in $^{29}$Si studies [24,27]. The topology of the silica structural frameworks in opal-CT remain a matter for debate, and the changes we observed between "simple" and "complex" opal-CTs may represent different structural states rather than different structural intergrowths.

### 4.5. Comments on Opal Formation and Transitions Between Opal-A, Opal-CT, Opal-C and Quartz

A recent paper [39] has proposed that temperature of formation ($\leq 45°$ for opal-AG and $>160°$ for opal-CT) is the prime determinant of opal type rather than type of deposit, i.e., volcanic versus

sedimentary [40,49] sources for opal-CT and opal-A, respectively. Our results are consistent with this proposition, our simple opal-CT showing a play-of-colour all come from deposits associated with volcanism.

It is possible that the transition between the different forms of opal derives from a similar process as the initial formation: for example, opal-A dissolution (partially hydrated silica to silicic acid) followed by deposition of opal-CT [41]. Analysis of dated sinter samples from hot springs sites [42] show the presence of opals with XRD patterns consistent with opal-A, a transitional form between opal-A and opal-CT, opal-CT, opal-C and quartz. The SEM images also show a change in form. While the opal-CT sample was probably consistent with "simple" opal-CT, verification is difficult owing to a significant amount of quartz. The SEM evidence for transition is also noted for geysers [38]. Changes in XRD pattern and near-IR have also been noted in accelerated aging at 300 °C of deep sea deposits [43].

We see two types of transitional forms that could be interpreted as opal-A to opal-CT and "simple" opal-CT to opal-C. However, these transitional samples are very uncommon and possibly far less than would be expected if this was a routine occurrence, but this could depend on the kinetics of the process and the geological age of the samples. The transformation of one form of opal to another is most probably a dissolution/reprecipitation reaction, given that opal is associated with flow of aqueous crustal fluids and these processes are known to be relatively rapid in terms of geological time [66,67].

Table 2 gives a brief summary of the defining characteristics found for the different opal types described in this work. The presence of "impurities" may cause misidentification for single samples.

**Table 2.** Summary of differentiating opal properties (this work).

| Opal Type | XRD | Raman [a] | IR [b] (ATR) | $^{29}$Si NMR [b] (Single Pulse) |
|---|---|---|---|---|
| Opal-AG | Very broad peak between ~2.2 Å and ~6.5 Å with maximum at 3.9–4.0 Å | Broad peak between ~230 and ~530 cm$^{-1}$ with maximum at ~370 cm$^{-1}$; 760–860 cm$^{-1}$ 970–975 cm$^{-1}$ | Peak at 530 cm$^{-1}$ | Q$_4$ FWHM 8.5 ppm Q$_3$ peak(s) not prominent |
| Opal-AN (hyalite) | Very broad peak between ~2.2 Å and ~6.5 Å with maximum at 3.9–4.0 Å | Broad peak between ~230 and ~530 cm$^{-1}$ with maximum at ~370 cm$^{-1}$ 760–860 cm$^{-1}$ 960–965 cm$^{-1}$ | Peak at 530 cm$^{-1}$ | Q$_4$ FWHM 8.3 ppm Q$_3$ peak visible as a shoulder |
| Opal-CT [c] | All have peak at 2.50 Å. Simpler types have a single peak at 4.08 Å. More complex types also show a peak or shoulder at 4.28 Å and a shoulder at 3.89 Å | Broad peak between ~180 and ~500 cm$^{-1}$ with maximum at ~300 cm$^{-1}$ to more defined maxima at 220, 295, 340 and 410 cm$^{-1}$ | *Absence of peaks at 530 and 625 cm$^{-1}$* | Q$_4$ FWHM 6.5 ppm Q$_3$ peak visible |
| Opal-C | 4.04 Å and 2.50 Å | Sharp peaks at 107, 222 and 409 cm$^{-1}$ | Peaks at 300, 385, 470 and 625 cm$^{-1}$ | *No common feature* |

[a] Without baseline correction. [b] Only unique features are noted. [c] Trend discussed in text.

## 4.6. Summary

This study provides a classification of examples from many sites, both gem quality and other samples, and incorporates a number of techniques. The large number of spectra and XRD patterns presented here illustrate the range of opals that may be found under each heading.

XRD remains the primary analytic method of choice as all samples, including those from newer sources, can be readily classified as opal-A, opal-CT, opal-C or a reasonable case may be made for a transitional form. We note that mid-IR ATR spectroscopy also fulfils this role. The XRD patterns for Opal-CT exhibited a range of forms from quite simple patterns to more complex ones, but this range was a continuum. At the "simple" end, the Ethiopian POC occurred while the "complex" forms included the common opals. Play-of-colour opals may belong to either the opal-A or opal-CT groups. Thus, the terms "precious opal", "play-of-colour", "potch", "common opal" and "fire opal" are best treated with caution, possibly only to be used within the trade, rather than in scientific studies.

The large body of samples examined in this work has allowed us to identify exemplars or typical specimens for the various opal groups. These provide authentic references that could be used in provenance authentication, in comparing purity and for providing characterised samples for further studies, such as chemical or thermal modification. New samples can be compared against these established and well-characterised examples. They may also be used for other studies such as geological (e.g., volcanic versus sedimentary) settings or trace element content.

**Supplementary Materials:** The following are available online at http://www.mdpi.com/2075-163X/9/5/299/s1.

**Author Contributions:** Conceptualization, A.P. and N.J.C.; Data curation, N.J.C.; Formal analysis: N.J.C., J.R.G. and M.R.J.; Investigation, N.J.C., J.R.G. and M.R.J.; Methodology N.J.C., J.R.G. and M.R.J.; Validation, N.J.C., J.R.G., M.R.J. and A.P.; Visualization, N.J.C. and A.P.; Writing—original draft preparation, N.J.C.; Writing—review and preparation, N.J.C., J.R.G. and A.P.

**Funding:** This research received no external funding.

**Acknowledgments:** The authors thank Ben McHenry of the South Australian Museum and Tony Milne of the Tate collection at the University of Adelaide for their unstinting assistance in locating samples for this study. The collection mangers of the Flinders University of South Australia, Museum Victoria and the Smithsonian National Museum of Natural History are thanked for the provision of samples and useful conversations. The authors acknowledge the expertise, equipment and support provided by Microscopy Australia and the Australian National Fabrication Facility (ANFF) at the South Australian nodes under the National Collaborative Research Infrastructure Strategy. We acknowledge access to the facilities at the Australian Synchrotron and the technical support provided by Dominique Appadoo for the Far-IR data collection. The constructive comments of the two anonymous referees and the associate editor are gratefully acknowledged.

**Conflicts of Interest:** The authors declare no conflict of interest.

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
