# Peer review of "A Review of the Classification of Opal with Reference to Recent New Localities"

_minerals, doi:10.3390/min9050299_

Round 1
Reviewer 1 Report
The work is surely intersting but I have some comments and critics.
In general I think the english form is not very accurate and the whole work should be revised from this point of view.
The introduction is too short and I think the authors should have to better explain the term of C, CT and A opals according with the classification by Jones & Segnit (1971) and others.
The authors assume that the only impurity in their samples is represented by quartz but in our experience in most of C and CT opals the most important impurity is represented by clays like smectite, saponite and others. The presence of clays also determines the purity and diaphaneity of the opals. How can the author be sure that there are no clays in their sample?? On the base of XRD analyses they could not determine this because they did not analise the range that comprises the peaks for clays. They should have to investigate the presence of clays in their samples.
Regarding the Figure 5, in our works we have alysed several precious CT opals from Tanzania, Peru and we have verified that their XRD patterns were similarv to those classified as complex opals while, according to the description of the authors, they should fall in the field of simple opals. So we think the classification reported in Fig. 5 is questionable. On our opinion do not really exist a marked difference between simple and complex CT opals but the passage is progressive.. Also, on the base of SEM observation is there a clear difference between simple and complex opals?
other corrections are the following:
Line 159: its "Haneti" and not "Hameti"
Line 164 and 165: the sentence that starts with "We note…." is not clear
Linev 166: the term "exemplar" is not very clear to me.
Line 206: I cannot really see this band in the figure, it is very weak or unexisting.
Line 246: it is not clear if you refer to the samples from Mezezo or o from Afar.
Line 258-260: please explain better as the sentence is not clear.
Line 289: the sentence "in baseline either side of the major pek" is not clear.
Lines 339 aqnd 340:I do not understand which figure I should have to see.
Line 476: add references.
The sentence comprised between the lines 489-491 is not absolutely clear.
Line 525 You speak of "Both hypothesis" but I cannot really understand what they are.
Line 394: its Ethiopia not Eithopia.
Line 410 and 413; why do you put "will"??
Line 436: its better to write various degrees of order and not structure.
In the paragraph 3.4.1 authors shoud have to insert the numbers of figures.
Figure 9: on my opinion the sample OOC4, also on the base of the SEM figure, is amorphous and not a CT.
Author Response
Referee 1
The work is surely intersting but I have some comments and critics.
In general I think the english form is not very accurate and the whole work
should be revised from this point of view.
We have gone through and tidied up the expression. We have endeavoured to explain some of the English words that the reviewer seemed unfamiliar with – such as exemplar (which mean typical)
The introduction is too short and I think the authors should have to better
explain the term of C, CT and A opals according with the classification by
Jones & Segnit (1971) and others.
We have expanded this section of the introduction and added a paragraph which explains the origin of the structural ideas behind the idea that opal-CT is an disordered intergrowth of cristobalite- and tridymite-like layers.
The authors assume that the only impurity in their samples is represented by
quartz but in our experience in most of C and CT opals the most important
impurity is represented by clays like smectite, saponite and others. The
presence of clays also determines the purity and diaphaneity of the opals.
How can the author be sure that there are no clays in their sample?? On the
base of XRD analyses they could not determine this because they did not
analise the range that comprises the peaks for clays. They should have to
investigate the presence of clays in their samples.
We did not the minor occurrence of clays and layer silicates in some of our samples. Those were their presence was significant we excluded from the detailed study
Regarding the Figure 5, in our works we have alysed several precious CT
opals from Tanzania, Peru and we have verified that their XRD patterns were
similarv to those classified as complex opals while, according to the
description of the authors, they should fall in the field of simple opals.
Section now recast to clarify things. We have pointed out the you might get simple and complex types from the same locality
We have expanded our coverage of POC opals
So we
think the classification reported in Fig. 5 is questionable. On our opinion do
not really exist a marked difference between simple and complex CT opals
but the passage is progressive..
This may turn out to be true but we think it is worth consideration. The novice user may however be perplexed by the range of XRD patterns and we can help with that.
We agree it is a continuum but opal-CT is a generic term that can cover a lot of things. We find that the notion of “simple” opal-CT is a convenient one. We have inserted the correlation with Raman to show there are discrete zones of opal-CT. but drawn back from fixing an arbitrary division. Simple and complex opal-CT no longer separated in the discussion
Also, on the base of SEM observation is
there a clear difference between simple and complex opals?
We don’t believe SEM is any more than indicative and it is certainly not quantitative. This would be a whole new study. We did an extensive amount of SEM on powdered samples originally, but cut nearly all the data was removed from the final draft, as it added little new that wasn’t already covered expensively by the literature
other corrections are the following:
Line 159: its "Haneti" and not "Hameti"
Line 164 and 165: the sentence that starts with "We note…." is not clear
Linev 166: the term "exemplar" is not very clear to me.
Line 206: I cannot really see this band in the figure, it is very weak or
unexisting.
Line 246: it is not clear if you refer to the samples from Mezezo or o from
Afar.
Line 258-260: please explain better as the sentence is not clear.
Line 289: the sentence "in baseline either side of the major pek" is not clear.
Lines 339 aqnd 340:I do not understand which figure I should have to see.
Line 476: add references.
The sentence comprised between the lines 489-491 is not absolutely clear.
Line 525 You speak of "Both hypothesis" but I cannot really understand what
they are.
Line 394: its Ethiopia not Eithopia.
Line 410 and 413; why do you put "will"??
Line 436: its better to write various degrees of order and not structure.
In the paragraph 3.4.1 authors shoud have to insert the numbers of figures.
This have all been corrected or superseded by other revisions
Figure 9: on my opinion the sample OOC4, also on the base of the SEM
figure, is amorphous and not a CT.
It’s an intermediate and has been described as such. SEM is not a primary classification method anyway. Our XRD and Raman data show that it is an intermediate between opal-A and opal-CT
This have been done or superseded.

Reviewer 2 Report
The paper "A review of the Classification of Opal with reference 2 to recent new localities " is an impressive and very extensive work on a wide set of opal. Authors propose a new way to classify opal-CT in sub-groups based on the diversity of opal-CT structure. The results of XRD, Raman and IR are very useful and may be considered as a good reference for future work on opal if the authors make all these data available as appendix (that I would strongly recommend). Figures of high quality are very clear. However, authors have some major point to elaborate.
Major issues :
1) Authors proposed to divide opal-CT into 2 sub-group ("simple" and "complex") but the strict limite between the two seems not clear. For example, the Figure 5 shows the spectrum of opal-CT, but the limit between complex and simple appear arbitrary. In fact, opal-CT is a continuous sequence (firstly established by Elzea and Rice 1996) and, in this context, proposing a strict limit appears irrelevant. Instead of this, I would propose to create an "index of complexity" consistent with these observations instead of a strict limit. For example, a geometrical criteria as the ratio of the peaks (area of heigh, 4.1 and 4.28 A) or a concavity measurements of the high angle shoulder would be more relevant.
2) The Appendix was not available to see the sampling. Few information are required to the overall sampling (how many of each type is the minimum, how many precious ...) in teh first pragraph of Materials and Methods. In the exemplar samples presented Table 1, among 50 opals, only 10 opal-A and 1 opal-C (so 39 oapl-CT, almost 80 %), is this representative of the overall sampling ? This fact is not discussed by authors. Do the over-representation of opal-CT increase the variability seen ? Perhaps with enough opal-A samples, a variability in structure may also be underlined.
3) The discussion requires a development and have serious flaws. The first part "Applicability of XRD for primary classification" is irrelevant as the classification of opal by Jones and Segnit is based on XRD, the applicability is thus obvious. Same for the second part that lists the spectrscopic methods used, but all the technics have already demonstrated their weaknesses and advantages for opal classification. If author want to propose their sub-classification, they need to demonstrate the applicatibility of their classification with these technics and not the applicability on the overall classification already established. The last part on opal formation is too superficial. The first statement lacks of discussion : opal-A may form at high temperature (geyrserite) and opal-CT may form in low-temperature (Wegel-Tena, Ethiopia). All this part need to be elabrotaed with relevant references.
4) Why scientist interested by opal need the new classification proposed by the authors ? This question need to be answered in the introduction. The "nomenclature problem" stated line 530 is not explained. Why there is a problem to have 2 classification (gemological and structural) ?
Minor remarks are included in the commented pdf, with a list of references to be included (with doi). A lot of sentences lacks commas, or some spaces (between the value and the unit), authors need to look at these.
Author Response
Referee 2
The paper "A review of the Classification of Opal with reference 2 to recent
new localities " is an impressive and very extensive work on a wide set of
opal.
Authors propose a new way to classify opal-CT in sub-groups based on
the diversity of opal-CT structure. The results of XRD, Raman and IR are very
useful and may be considered as a good reference for future work on opal if
the authors make all these data available as appendix (that I would strongly
recommend).
We talk a bit about a growing corpus of results and using established a well characterised samples set of samples for future work. (In the coming weeks we have some inelastic neutron scattering experiments and more detailed NMR studies on a subset of samples so see if this clarifies that nature of the CT structure
We make the point that there may be variety in samples taken for the same place so a compendium should be taken with a grain of salt
Figures of high quality are very clear. However, authors have
some major point to elaborate.
Major issues :
1) Authors proposed to divide opal-CT into 2 sub-group ("simple" and
"complex") but the strict limite between the two seems not clear. For example,
the Figure 5 shows the spectrum of opal-CT, but the limit between complex
and simple appear arbitrary. In fact, opal-CT is a continuous sequence (firstly
established by Elzea and Rice 1996) and, in this context, proposing a strict
limit appears irrelevant. Instead of this, I would propose to create an "index of
complexity" consistent with these observations instead of a strict limit. For
example, a geometrical criteria as the ratio of the peaks (area of heigh, 4.1
and 4.28 A) or a concavity measurements of the high angle shoulder would
be more relevant.
Section on simple and complex recast as we have taken on board the referees’ comments. We have downplayed a formal division and noted two extremes with a range in between.
Agree we have verified the continuous sequence but in this case we have provided 3 versions of it: P4 width and appearance of the P1-P3 complex, P1 v P3 ratio correlation and P1 v P4 FWHM correlation. We have also shown a correlation with Raman. Our approach has been more empirical – finding measurable data, presenting and analysing it. This may be contrasted with assuming a mix of cristobalite and tridymite and using this to explain results.
We have added a section on “index of complexity” and why it is challenging. It is probable that there aren’t two extremes anyway and that the two or more axes will be needed for grouping. We noted that Fig5 is based on the FWHM of peak 4 and it might not be the only way of classifying things. We are continuing to look at this but haven’t found anything based on chemistry/crystal chemistry.
The Raman spectra show some of the problem as the opal-CT samples have different spectra. We have inserted a figure to show a correlation between the XRD patterns and Raman spectra.
Interestingly we find no correlation with the occurrence of POC
Simple and complex opal-CT no longer separated in the discussion
2) The Appendix was not available to see the sampling. Few information are
required to the overall sampling (how many of each type is the minimum, how
many precious ...) in teh first pragraph of Materials and Methods. In the
exemplar samples presented Table 1, among 50 opals, only 10 opal-A and 1
opal-C (so 39 oapl-CT, almost 80 %), is this representative of the overall
sampling ? This fact is not discussed by authors. Do the over-representation
of opal-CT increase the variability seen ? Perhaps with enough opal-A
samples, a variability in structure may also be underlined.
Some figures have been added and the analysis was more comprehensive than may have been implied. The appendix will be sent this time. We have explained that table 1 only represents the highlighted samples with the rest in the appendix, particularly for newer samples
3) The discussion requires a development and have serious flaws. The first
part "Applicability of XRD for primary classification" is irrelevant as the
classification of opal by Jones and Segnit is based on XRD, the applicability
is thus obvious.
We have shown it still holds for the newer sites (ie newer than 1971). This was important as POC opals were being found that are not opal-A. We have expanded the visibility and scope of the Jones and Segnit classification so that the record contains more examples.
Same for the second part that lists the spectrscopic methods
used, but all the technics have already demonstrated their weaknesses and
advantages for opal classification. If author want to propose their subclassification,
they need to demonstrate the applicatibility of their
classification with these technics and not the applicability on the overall
classification already established.
We think this is mostly about simple and complex which we think we have addressed. We have noted that there are distinct zones in the opal-CT continuum and this should be recognised.
Any particular opal-CT can be placed it within the established bounds
We are however reluctant to comment on the nature of the trends along the lines in the diagrams shown as we have no (as yet) structural or chemical evidence. For instance, does an opal move from one end to the other as it “ages” or is the position due to the formation conditions? Our work may provide a reference for such studies.
The last part on opal formation is too
superficial. The first statement lacks of discussion : opal-A may form at high
temperature (geyrserite) and opal-CT may form in low-temperature (Wegel-
Tena, Ethiopia). All this part need to be elabrotaed with relevant references.
References added. This was included to provide context to the paper. It’s a lead in to any potential work on diagenesis (as above), which is currently being undertaken by other collaborators using some of our exemplar samples.
4) Why scientist interested by opal need the new classification proposed by
the authors ? This question need to be answered in the introduction.
We just wanted to examine homogeneity and found that there is an issue in the opal-CT XRD patterns and it can be resolved to some extent by curve fitting. A chemist would be concerned by the range of “acceptable” patterns for opal-CT. We can confidently say where a specific opal-CT lies in the continuum. Further studies may provide more information and it may allow reinterpretation of previous studies. For instance, attempts at curve fitting of Raman spectra may be revisited in terms of the XRD pattern (is it a generic observation?). We also hint that NMR may provide something.
The
"nomenclature problem" stated line 530 is not explained. Why there is a
problem to have 2 classification (gemological and structural) ?
This was done to avoid the “them and us”. It does however red flag that something peculiar is going on when there were many years of only precious opal-A where the POC was explained by something unique. It’s different now. Ethiopian samples have accentuated this.
Minor remarks are included in the commented pdf, with a list of references to
be included (with doi). A lot of sentences lacks commas, or some spaces
(between the value and the unit), authors need to look at these.
This have been done or superseded.
Round 2
Reviewer 1 Report
The authors followed our suggestions and the work is surely improved.
Author Response
- Far and Medium Raman spectroscopy should be corrected and simply mention it as
Raman spectroscopy throughout the text.
We have made several corrections throughout the text regarding this issue.
- Page 3 second paragraph: The laser excitation is usually 785 nm and not 786 nm;
please confirm or change it accordingly.
The wavelength used was measured to be 785.63 nm. We had chosen to round up to 786nm. However, our manuscript incorrectly stated the red laser to be 638nm, when it was actually measured to be 639.82nm, thus we have adjusted the text to state 640nm for this laser.
- Add the slit width use to acquire Raman spectra or the spectra resolution.
The slit width and grooves/mm of the grating are irrelevant without further information regarding the spectrometer. We have thus given a FWHM value for the resolution and also removed the groves/mm value.
- At line 113 of page 3 is mentioned a laser power of 27mW for 786 nm; is this the
full laser power or the laser power on the sample after passing through all the filters and lenses? Add the laser power on the sample used for the measurements for all the lasers.
We have now stated that the power is measured at the sample and have added laser powers for all the lasers used.
- Add how the calibration of the Raman instrument was performed (silicon or diamond)
and at which position (520.6 cm-1 for silicon/1331.8 cm-1 for diamond or different?).
We have added that the instrument was calibrated using the silicon peak.
- Page 3 line 123: Write wavenumber instead of wavelength
We have added this and a second occurrence in the text.
- Page 16 line 426: Write Mezezo instead of Mezozo
We have corrected this.
- Page 21 line 599, Table 2: Add the FWHM of all the peaks (even a range) and avoid
to mention very broad etc. Some of the peaks needs decompositions/deconvolution but this is a lot of work more; simple mention the ranges of FWHM with a note that these peaks might consist of several bands.
We have mentioned in the manuscript the difficulty in deconvolution the Raman spectrum because of unknown baseline. We have included ranges instead of FWHM for Raman and XRD, but retained the descriptive terms very broad etc. since FWHM is not a differentiating feature.
The FWHM is however a differentiating feature for NMR so we have retained it.
- Some criteria was established using NIR spectroscopy but it is not mentioned in
the article; please take in account the article below and add some sentences in the discussion/perspectives regarding that:
https://www.schweizerbart.de/papers/ejm/detail/29/87623/Near_infrared_signature_of_opal_and_chalcedony_as_a_proxy_for_their_structure_and_formation_conditions
This reference has been added to other NIR papers that we had already included in the manuscript.